



# Examining Spin-Up Behaviour within WRF Dynamical Downscaling Applications

Megan S. Mallard[1], Tanya L. Spero[1], Jared H. Bowden[2], Jeff Willison[1], Christopher G. Nolte[1], Anna M.
Jalowska[1]

[1]Office of Research and Development, U.S. Environmental Protection Agency, Research Triangle Park, North Carolina,
USA
[2]North Carolina State Climate Office, Raleigh, North Carolina, USA

*Correspondence to*: Megan S. Mallard (mallard.megan@epa.gov)

**Abstract.** Spin-up is the period after initialization when a model transitions away from its dependence on initial conditions toward a dynamic equilibrium between driving boundary conditions and its own internal dynamics. Regional climate models (RCMs) are often used to simulate conditions over several decades to inform local adaptation and resilience activities. The spin-up period represents added cost to already resource-intensive simulations, and it is often infeasible to use a spin-up period that produces complete model equilibrium. Therefore, a pragmatic compromise is desired to minimize the effects of

spin-up. Here, two overlapping dynamically downscaled simulations using the Weather Research and Forecasting model over the contiguous U.S. (31-year and 11-year integrations) are used to explore convergence associated with model spin up. The shorter simulation is initialized 20 years after initialization of the longer (reference) run, and the runs are analysed over the period covered by both simulations, giving the reference simulation a 20-year period to attain spin-up prior to comparison. After initialization, the shorter run features cooler surface and near-surface temperatures and greater soil

moisture compared to the reference simulation. Differences between the runs decrease in magnitude over the first 3 months as autumn transitions to winter; however, these differences re-emerge and reach a secondary peak during the proceeding spring and summer. During this warm season, evaporation and accompanying evaporative cooling increase and temperature differences between the simulations re-emerge. These results support using at least one year of spin-up time in RCM applications to account for the seasonality of spin-up behaviour.

**1 Introduction**

Regional climate modelling, or dynamical downscaling, applications provide data to inform adaptation and resiliency at local and community scales, including extremes in near-surface temperature or precipitation. One benefit of using dynamical downscaling (as opposed to statistical downscaling) is that a dynamically consistent suite of three-dimensional (3D) fields is created at sufficient temporal frequency to drive other environmental models, such as air quality or hydrology models.

Generating credible dynamical downscaling outputs can present several challenges to serve a variety of applications while remaining computationally manageable.



In general, a spin-up period follows model initialization to allow the model solution to transition from being strongly influenced by the initial condition to a state of dynamic equilibrium between the model physics and boundary forcing from the driving dataset (Georgi and Mearns, 1999; Denis et al., 2002). Therefore, the spin-up is a period over which the model
initial conditions are "forgotten" as information from the lateral boundaries and the model's internal physics generate this equilibrium. Long-running simulations (such as those generated by dynamical downscaling) with the same driving conditions should converge towards more similar results as the influence of their initial conditions lessens over time.

In dynamical downscaling, spin-up time can be influenced by several factors, such as the size of the model domain (Leduc and Laprise, 2009), physics configuration (e.g., Jankov et al., 2007; Kleczek et al., 2014; Tewari et al., 2022), or including
erroneous or poorly resolved features in initial conditions (e.g., Cosgrove et al., 2003; Jerez et al., 2020). Spin-up periods may also be chosen to ensure the initialization captures antecedent conditions or steering flow within a case study (e.g., Denis et al., 2002; Liu et al., 2023). Selecting a spin-up period can be critical to minimize the effects of spurious behaviour within regional climate model (RCM) simulations that are run "in parallel" (utilizing multiple initializations to break the simulated period into several segments), resulting in multiple spin-up periods throughout the simulation (e.g., Lavin-Gullon
et al., 2023; Rahimi et al., 2024).

Jerez et al. (2020) examined spin-up duration and the influence of seasonality using the Weather Research and Forecasting (WRF) model as an RCM for a European domain with 50-km grid spacing, with driving conditions provided by a global climate model (GCM) from the 5[th] Coupled Model Intercomparison Project (CMIP5) (Taylor et al., 2012). Simulations of various lengths were compared to a reference simulation that was initialized 2 years prior and both atmospheric and soil
fields were included in the analysis. Although Jerez et al. concluded that 2-m temperature and precipitation were spun up after one week, a 6-month spin-up period was recommended as a general guideline for RCM applications when assessing optimal model performance across both atmospheric and soil conditions. They note that longer spin up periods can be required when key physical mechanisms considered in the application are dependent on soil conditions, as longer times are needed for soil conditions to converge towards the reference simulation. However, Jerez et al. concluded that even one
year—the longest period considered in the study–was insufficient for deep soil moisture to approach equilibrium.

Cosgrove et al. (2003) conducted a spin-up study focused on soil conditions within the North American Land Data Assimilation System (NLDAS, spatial resolution of 0.125°) utilizing 4 different land surface models (LSMs) across various experiments, which were driven by external meteorological forcing. They found multi-year spin-up times were needed for soil conditions in each LSM, including the Noah LSM. Cosgrove et al. performed recursive experiments with the same
driving data used over 11 annual cycles to give several years for the LSM to approach equilibrium with the driving conditions. They found that the soil fields could require more than 10 years to achieve a "fine-scale equilibrium" over their domain covering the contiguous U.S. (CONUS). For that study, the timing for this equilibrium was assessed by comparing the same month in each of the annual cycles and equilibrium was achieved when the percentage change compared to the last annual cycle was less than 0.01%.



Cosgrove et al. highlighted the influence of the physical processes in the LSM that are utilized to "forget" erroneous conditions in the initial state to achieve spin-up of land-surface conditions. They compared a reanalysis-driven LSM simulation to idealized experimental runs where soil moisture was represented by either idealized "dry" and "wet" initial conditions. The wet run converged toward the reanalysis-driven simulation relatively quickly, with the control and wet simulations achieving a "practical" equilibrium (percentage change < 1%) at ~1-2 years, respectively. Meanwhile, the dry

run took an additional 3-4 years (in comparison to the control simulation) to achieve this metric of spin-up. Cosgrove et al. concluded that the wet simulation could immediately spur evaporation to reduce high saturation deficits, but the drier run required sufficient precipitation to moisten the soil and converge toward the reference solution. Therefore, longer spin-up times can be expected when the model relies on intermittent dynamical processes (such as precipitation) to reduce the influence of its initial conditions over time.

Here, spin-up time is examined using simulations from the EPA Dynamically Downscaled Ensemble (EDDE) Version 1 dataset (Nolte et al., 2021; EPA, 2024, 2025; Spero et al., 2025). EDDE contains dynamically downscaled projections of various CMIP5 GCMs using the WRF model in the historical past and under various future scenarios (van Vuuren et al., 2011). While early work testing an experimental set-up for EDDE used spin-up periods of 1-2 months (Otte et al., 2012; Bowden et al., 2012), the EDDE Version 1 methodology employed a 3-month spin-up period (Nolte et al., 2018; Nolte et al.,

2021). These spin-up periods are comparable to those used by other dynamical downscaling applications. However, this aspect of the methodologies employed for regional climate and dynamically downscaled simulations vary widely. Representative examples range from the 1-month spin-up periods employed within the Western US Dynamically Downscaled Dataset (WUS-D3) dataset (Rahimi et al., 2024) to the 1-year of spin-up recommended by guidance within the COordinated Regional climate Downscaling EXperiment (CORDEX, 2021). Choices in spin-up among these applications

can be determined based on which physical processes and affected outputs are considered most important for their respective applications, the physics of the underlying RCM being used, and compromises imposed by limited computational resources that constrain the total length of a simulation.

Here, to assess the effects of spin-up behaviour using the EDDE data, two EDDE historical simulations that are initialized 20 years apart are analysed over an 11-year overlapping period. This framework facilitates analysis of the spin-up of

atmospheric variables and soil conditions by comparison against a reference simulation that has two decades to spin up. This analysis advances upon prior studies that used sub-decadal timescales to examine spin-up of a full 3D RCM.

## 2 Methods

Two RCM simulations are driven by the Community Earth System Model (CESM, archived resolution of 0.9° × 1.25°) (Gent et al., 2011) for overlapping historical periods using version 3.4.1 of the WRF model (Skamarock and Klemp, 2008)

on a 36-km domain (Fig. 1). The simulations used 34 terrain-following layers extending to a model top at 50 hPa. Both runs are initialized at 00 UTC on 1 October, with simulation "I74" initialized in 1974 and "I94" initialized in 1994. Both I74 and



I94 are continuous integrations with only a single initialization. Both runs end at 00 UTC on 1 January 2006, such that I74 covers a 31-year and 3-month period, while I94 covers the last 11 years and 3 months of I74. I94 has been used to examine the impacts of climate change on air quality (Fann et al., 2015; Nolte et al., 2018, 2021) and phenological indicators (Mallard et al., 2023). I74 was used to simulate extreme rainfall events to create rainfall intensity-duration-frequency curves (Jalowska et al., 2021).

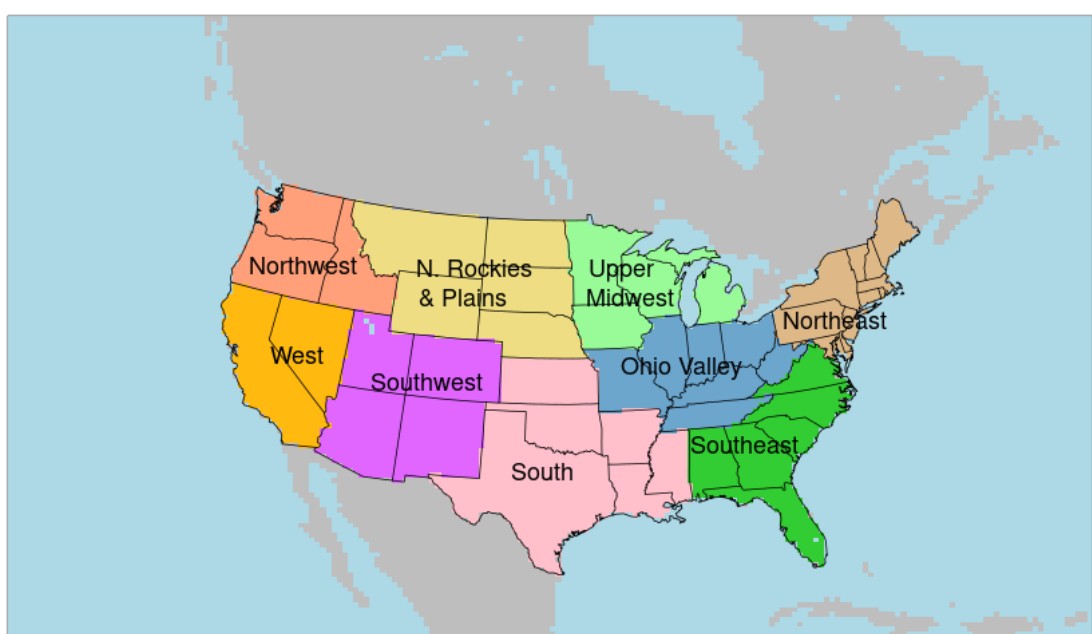

**Figure 1. The 36-km WRF domain used for simulations I74 and I94, with the nine National Centers for Environmental Information (NCEI) regions shown within the CONUS.**

Both WRF runs use matching physics configurations including the Kain-Fritsch scheme to parameterize convection (Kain, 2004) with radiative feedback following Herwehe et al. (2014), WSM6 microphysics (Hong and Lim, 2006), the Yonsei University planetary boundary layer (PBL) scheme (Hong et al., 2006), and the Rapid Radiative Transfer Model for global models (Iacono et al., 2008) for longwave and shortwave radiation. Spectral nudging (Miguez-Macho et al., 2004) is applied to geopotential heights, horizontal wind components, and temperature following Otte et al. (2012).

Although spectral nudging may influence spin-up behaviour (Gómez and Miguez-Macho, 2017), nudging is applied here only to large-scale atmospheric features above the PBL and not to specific humidity. Near-surface vapor pressure deficit is allowed to evolve without direct influence from spectral nudging. Gómez and Miguez-Macho (2017) demonstrated the



influence of nudging on spin-up time and concluded that the simulation with spectral nudging (similarly, applied above the PBL) had a faster spin-up time (at ~36-48 hours) relative to a simulation with no nudging applied (at 96 hours). Meanwhile, a grid nudged simulation was found to reach spin-up the fastest of all runs. The authors concluded that this most constrained simulation (with nudging applied across the entire energy spectra) allowed the grid nudged simulation to reach a balance between the nudging influence and its own internal dynamics relatively quickly; meanwhile, the simulation without nudging had the least constraint and the longest spin-up time. Therefore, while nudging has been found to effect spin-up, these prior results suggest its influence is found at temporal scales of hours to a few days and not at the longer timescales more typical of spin-up in RCM applications.

The Noah LSM (Chen and Dudhia, 2001) and the Revised MM5 Monin-Obukhov surface scheme (Jimenez et al., 2012) are used to simulate surface and soil processes. The Noah LSM has 4 soil layers (0-10, 10-40, 40-100, and 100-200 cm, respectively, from top to bottom), and gravitational flow is allowed out of the bottom layer. Monthly average soil temperature and moisture from CESM provide initial soil conditions for WRF. The 24-category U. S. Geological Survey (USGS) land use dataset provides landuse information (Loveland et al., 2000), and lake surface temperatures are incorporated from the CESM Community Land Model as described in Spero et al. (2016).

## 3 Results

I74 and I94 are compared using several atmospheric and soil fields during the overlap in their simulation periods. In this analysis, I74—which was initialized 20 years prior to the start of I94—serves as the reference simulation. It is assumed that the atmospheric and soil fields in I74 are in equilibrium prior to the initialization of I94. The analysis period begins at 00 UTC 1 October 1994 (the initialization time for I94) and ends at 00 UTC 1 January 2006.

### 3.1 2-m Temperature

At initialization in October 1994, the CONUS-wide average 2-m temperature from I94 is ~0.4 K cooler than I74 (Figs. 2a & 2b). Although the averaged difference between the runs diminishes rapidly from ~0.4 K in magnitude to <0.1 K in the first 3 months after initialization, it diverges the following spring and forms a secondary peak in magnitude of ~-0.15 K during the summer of 1995 (Fig. 2b). Seasonally averaged differences in 2-m temperature are shown for the first winter (December 1994–February 1995) and first summer (June–August 1995) of I94, compared with I74 (Figs. 2c & 2d, respectively). During the first winter, I94 is notably cooler than I74 in the South [specifically, the states of Texas, Louisiana, Mississippi, and Arkansas; which are located in the southeastern portion of the South NCEI region (Fig. 1)]. By the following summer, I94 has cooler temperatures than I74 across portions of the South, Ohio Valley, Upper Midwest, and Northern Rockies and Plains regions (Fig., 2d), while CONUS-averaged differences reaching a secondary low of ~-0.15 K (Fig. 2b). Although a seasonal cycle of convergence followed by a re-emergence of temperature differences occurs during this first year of the timeseries, from the beginning of 1996 and onward, the monthly and spatially averaged 2-m temperatures differ by less than ±0.05 K over the rest of the decadal period.





**Figure 2. A timeseries beginning in January 1994 of monthly and spatially averaged 2-m temperature over the CONUS for I74 (blue) and I94 (red) (panel a) and the difference between those timeseries beginning in October 1994 (I94 minus I74; panel b). The difference (I94 minus I74) in seasonally averaged 2-m temperature [K] for the first**
**winter (panel c, December 1994–February 1995) and summer (panel d, June–August 1995) following initialization.**





### 3.2 Soil Temperature and Moisture

A seasonal cycle is also apparent in top-layer (0-10 cm) soil temperatures (Figs. 3a and 3b). Over the CONUS, soil temperature shortly after initialization is colder in I94 than I74 for both top-layer (Figs. 3c and 3d) and lower-layer soil temperatures (not shown). The magnitude of the monthly and spatially averaged differences in top-layer soil temperature

exceeds those of 2-m temperatures over the CONUS but follows a similar pattern of convergence (decreasing in magnitude) over the initial transitional season, followed by a divergence during the subsequent spring and summer where a secondary peak in the magnitude of difference between the two simulations occurs. During summer 1995, differences in soil temperature are largest in areas of the central CONUS, corresponding to differences in 2-m temperature (Figs. 2d and 3d). While 1994-1995 wintertime differences in soil temperature are relatively small over the CONUS (approximately ±1 K), the

differences are accentuated in the Canadian Prairies (Fig. 3c). Portions of the model domain in Canada exhibit substantially warmer temperatures within top and deeper layer (not shown) soil temperatures during this period. Land-surface processes in frozen soils and snow cover may impact spin-up in those areas.





**Figure 3. As in Fig. 2 but for top-level (0-10 cm) soil temperature [K].**


Alongside cooler temperatures at the surface and near-surface in the I94 initialization relative to I74, generally soil moisture values are also wetter in the top layer (Fig. 4) and deeper layers (Fig. 5). A timeseries of monthly-average top-layer soil moisture (Figs. 4a and 4b) shows that most of the convergence between the two simulations occurs within the first 15 months as soil moisture generally decreases in I94 relative to I74. Over the first winter following initialization, wet and dry



biases (in I94 compared against I74) are present across the CONUS but greater soil moisture is found throughout much of the central U.S. (Fig. 4c). Similar to surface and near-surface temperatures, CONUS-average top-layer soil moisture exhibits a seasonally influenced cycle of rapid convergence over the first few months followed by divergence and a secondary peak in summer 1995 when wet biases are increased throughout the central CONUS (Figs. 4b & 4d).

![Figure 4 panels a) time series of monthly average top-level soil moisture, b) difference, c) Winter 1994 map, d) Summer 1995 map]

**Figure 4. As in Fig. 2 but for top-layer (0-10 cm) soil moisture expressed as water fraction by volume [m³ m⁻³].**



Differences in mid- and bottom-layer soil moisture between I74 and I94 persist longer (Figs. 5a and 5b) than differences in 2-m temperature and top-layer soil temperature and moisture. Prior studies have found that deep soil moisture requires longer spin-up durations than other LSM variables, spanning years to decadal periods (e.g., Cosgrove, 2003; Jerez et al., 2020). Also, unlike the top-layer soil temperature and moisture, a seasonal cycle is not apparent over the 1994-1995 period in lower-layer soil moisture (Fig. 5). Instead, higher mid- and deep-soil moisture over the CONUS in I94 decreases precipitously towards I74 throughout 1995 and finally converges in the summer of 1996, with minimal CONUS-averaged differences over the rest of the simulation (Figs. 5a & 5b).





**Figure 5. As in Fig. 2 but for lower-layer soil moisture [m³ m⁻³] from three layers aggregated over 10–200 cm using a depth-weighted average. Note that, for brevity, this study aggregates the soil fields shown in Fig. 5 by using weighted average (by depth) soil moisture over the bottom 3 soil layers.**





### 3.3 Latent and Sensible Heat Flux

The I94 simulation has a soil state that is wet and cool relative to I74 throughout areas of the central CONUS (Figs. 3 & 4).
In WRF, the LSM interacts with the overlying atmosphere via turbulent fluxes of heat and moisture. In the months following initialization, latent heat flux in I94 (Figs. 6a and 6b) is higher than in I74, while sensible heat flux (Figs. 7a and 7b) is reduced relative to I74. This is physically consistent with a top-layer soil state with greater moisture and cooler temperatures. The timeseries of monthly and spatially averaged surface fluxes in I94 converge towards I74 over the first ~3 months of simulation time, with differences in fluxes most apparent only in the South during winter 1994 (Figs. 6b & 6c).
During the following summer, a secondary peak in increased latent heat flux in I94 occurs in the timeseries, as evaporation increases within I94 relative to I74 across the central CONUS, extending from the South through the Ohio Valley, Upper Midwest, and Northern Rockies and Plains (Fig. 6d). These areas of the central CONUS that exhibit the largest differences in surface and near-surface temperature and moisture are dominated by various forest, cropland, and pasture landuse types, as shown by Mallard et al. (2018; their Fig. 2) in similar 36-km WRF simulations utilizing the USGS landuse data. Meanwhile,
sensible heat flux is lower in I94 than in I74 over the central CONUS during the first summer of the simulation over the central CONUS (Fig. 7d), which is consistent with cooler top-layer soil temperatures in I94 (Fig. 3d).





**Figure 6. As in Fig. 2 but for latent heat flux [W m⁻²].**




**Figure 7. As in Fig. 2 but for sensible heat flux [W m⁻²].**

The seasonal variations in latent heat flux simulated by both I94 and I74 are driven by multiple factors. In the Noah LSM, evapotranspiration within a grid cell is the sum of evaporation from bare soil, evaporation from a wet canopy, and transpiration from plants (Chen and Dudhia, 2001). All three components are directly related to the rate of potential evaporation, $E_p$. The Noah LSM also includes inhibiting factors that can prevent evaporation from occurring at this rate, such





as vegetation wilting and the partitioning of evaporation (between bare soil and the canopy) based on the fraction of green vegetation (e.g., Chen and Dudhia, 2001; Ek et al., 2003; Chaney et al., 2016). The formulation of $E_p$ within the Noah LSM is based on the Penman approach implemented by Mahrt and Ek (1984), where $E_p$ is proportional to the saturation specific humidity of the overlying atmosphere, which is a function of near-surface atmospheric temperatures. As temperatures

decrease throughout the winter, saturation vapor pressure deficits decrease. At the same time, reduced leaf-area index (LAI) results in less transpiration out of the canopy, hampering evaporation within the grid cell. During summer, moisture demand and LAI increase, among other changes, resulting in the robust seasonal cycle of latent heat flux simulated by both I94 and I74 (Fig. 6a).

The relatively wet initial soil state in I94 (Figs. 4b and 5b) results in increased evaporation following initialization, but

differences between average soil moisture within I94 and I74 are suppressed over the first winter (3-5 months into the I94 simulation), when evaporation is suppressed by the seasonal influences. Meanwhile, mid- and lower-layer soil moisture remain high in I94 relative to I74 (Figs. 5b and 5c). During the following summer, the greater soil moisture in all soil layers, along with seasonally increased saturation deficits in the overlying atmosphere, supports further increases in evaporation and latent heat flux in the I94 simulation (relative to I74) over time (Figs. 4d, 5d, 6d). Evaporative cooling contributes to cooler

temperatures in I94 (compared with I74) over the warm season (Fig. 2d).

### 3.4 Quantitative Examination of Spin-up

Here, statistical criteria are used to examine spin-up times for several variables within each of the NCEI regions and over the CONUS (Table 1). For each variable, cumulative distribution functions (CDFs) are generated for each grid cell for daily-averaged fields from the I74 and I94 simulations and then compare them using the Kolmogorov-Smirnov (K-S) test (ks.test

in the stats package of R version 4.3.0) where p-value ≤ 0.05 determines statistical significance. A 1-year window is used to generate each CDF, beginning on the first day of the simulation, then the 1-year window is incremented day-by-day until the statistical criteria are met. Here, a 1-year window is used to assess whether statistical criteria have been met so that the potential seasonal re-emergence of differences between I74 and I94 is considered in each comparison. When the criteria are met, the start date for that 1-year window is considered the time at which spin-up period for that grid cell is sufficient.

Within each region (Fig. 1), the regional spin-up time is designated as the first date when most land grid cells (> 50%) in that region have achieved spin-up. This methodology is similar to spin-up criteria utilized by Jerez et al. (2020), where a K-S test was used to determine when fields from an experimental simulation were statistically similar to those in the reference simulation.

This spin-up criterion can assess whether the runs have converged sufficiently to produce fields with similar distributions

over an annual cycle, regardless of the timing of events. Because these CDFs include weather events throughout the year, similarly extreme events may be simulated in I74 and I94 but could be temporally displaced at a given grid point without affecting the statistical criteria for spin-up. Therefore, the comparison of CDFs provides a useful framework for examining





spin-up behaviour in RCM applications. This approach may be less appropriate for applications where incipient conditions and the timing of events are more important, such as case studies or pseudo global warming experiments.




|  | 2-m Temperature | Top-Layer Soil Temperature | Bottom-Layer Soil Temperature | Top-Layer Soil Moisture |
|---|---|---|---|---|
| **CONUS** **n=5936** | 1-365 d | 1-365 d | 10.8 | 21.1 |
| **Northwest** **n=489** | 1-365 d | 1-365 d | 4.7 | 3.8 |
| **West** **n=526** | 1-365 d | 1-365 d | 3.5 | 11.2 |
| **N. Rockies & Plains** **n=938** | 1-365 d | 5.7 | 22.7 | 50.6 |
| **Southwest** **n=841** | 1-365 d | 1-365 d | 10.1 | 15.1 |
| **Upper Midwest** **n=508** | 1-365 d | 4.4 | 16.9 | 42.5 |
| **South** **n=1114** | 1-365 d | 0.9 | 11.0 | 18.2 |
| **Ohio Valley** **n=608** | 1-365 d | 1-365 d | 11.0 | 32.6 |
| **Southeast** **n=566** | 1-365 d | 1-365 d | 10.8 | 18.5 |
| **Northeast** **n=346** | 1-365 d | 1-365 d | 4.9 | 12.0 |

**Table 1. Over all 9 NCEI regions and the CONUS, the number of months until spin-up is achieved (or listing as 1-365 days where spin-up is achieved during the first annual cycle tested) for variables discussed in the text, according to the criteria described in the text. The number of grid cells is also listed for each region and the CONUS.**



Spin-up for 2-m temperature and top-layer soil temperature occurs within the first annual window (days 1 through 365) in
most regions (Table 1). However, the Northern Rockies and Plains, the Upper Midwest, and the South take 0.9-5.7 months
to spin-up top-layer soil temperature. As expected, deeper soil temperatures require longer spin-up times, ranging between
3.5-22.7 months, with about 10.8 months sufficient to spin up across the CONUS. Top-layer soil moisture spin-up occurs by
21.1 months across the CONUS but varies widely regionally, with spin-up occuring as quickly as 3.8 months for the
Northwest but as slowly as 50.6 months in the Northern Rockies and Plains region. For bottom-layer soil moisture, the
criteria applied here did not result in a majority of grid cells meeting spin-up criteria over the CONUS and for several of the
NCEI regions. Bottom-layer soil moisture in the West and Northwest achieve spin-up at 37.3 and 55.0 months, respectively.
Generally, prolonged spin-up times are needed for areas throughout the central CONUS.

The Northern Rockies and Plains region has the longest spin-up times for all variables where regional differences are
apparent, while the Upper Midwest has the second longest. These areas of the CONUS exhibit the largest seasonally-
influenced differences in soil temperature and moisture in I94 relative to I74 (Figs. 3 & 4). Contrastingly, spin-up is
generally achieved more quickly in the western portion of the CONUS with the shortest spin-up times for deep soil
temperatures and moisture in the West, and for top-layer soil moisture in the Northwest. Here, confluence between the
increased soil moisture in I94 compared with I74 plays a key role in regional spin-up of soil moisture values. Based on the
idealized work of Cosgrove et al. (2003), it could be speculated that initially drier soil moisture values in the experimental
simulation could result in even longer spin-up time in regions where initial differences are the largest, as that study found
that an excessively wet simulation spun up more quickly than an idealized dry simulation, as described in Section 1. While 1
year satisfies spin-up of 2-m and top-layer soil temperatures, spin-up periods for top-layer soil moisture exceed 1 year for
most regions and over the CONUS. Therefore, while utilizing at least a year of spin-up time would mitigate the obvious
seasonal signals of the spin-up behaviour highlighted above, multiple years of spin-up (~1-4) may be needed in some regions
for spin-up of soil moisture (i.e., in this experiment, the Northern Rockies and Plains and the Upper Midwest).

## 4 Conclusions

Here, spin-up behaviour is examined in an RCM, where surface and near-surface fields in an 11-year simulation, I94, are
compared with those from a 31-year reference simulation, I74, that is initialized 20 years earlier. Each is driven by the same
CMIP5 GCM and uses the same RCM configuration, aside from initialization date and initial model state. Prior related
studies of spin-up periods in regional climate modelling have used "reference" or "control" simulations which predate their
experimental runs by as much as 2 years (Jerez et al., 2020) or less than 30 days (Pan et al., 1999). The current study uses a
multi-decadal period for the reference simulation, which better supports the implicit assumption that the reference run has
already undergone spin-up and that comparison with it can provide a robust analysis of spin-up behaviour in the
experimental simulation. This is especially important for assessing whether the soil state is spun-up, as soil processes in the





model have been found to require longer spin-up times than atmospheric processes (e.g., Cosgrove et al., 2003; Jerez et al., 2020; Lavin-Gullon et al., 2023).

Comparing I94 to the reference I74 shows that key surface and near-surface variables (2-m temperature and top-layer soil temperature and moisture) exhibit a seasonally influenced pattern as they converge over the first year of the simulation (Figs. 2-4). Generally, the initial state in I94 has wetter soil moisture values and cooler near-surface and soil temperatures than I74.

The model solutions artificially appear to converge over the first ~3 months from the October 1 initialization into the winter months. However, during the proceeding spring, differences between the two simulations re-emerge and reach a secondary peak the following summer.

I94 increases latent heat release and reduces sensible heat flux (Figs. 6 and 7) in response to cooler and wetter soils, following the same seasonal pattern of differences as 2-m temperature and top-layer soil temperature that are reduced in

magnitude over the winter and reappeared the following summer. Meanwhile, during the initial year of the I94 simulation, deeper soil moisture values remain elevated in I94 (Fig. 5), providing additional soil moisture to drive increases the following summer when seasonally warmer temperatures support higher saturation deficits and increased evaporation. Cooler temperatures in I94 relative to I74 also re-emerge during summer 1995 (the first summer of the I94 simulation), as more near-surface evaporative cooling in the I94 run sustains cooler surface temperatures relative to I74.

The full annual cycle should be considered when choosing optimal spin-up for RCM applications. If analysis had been limited to a 3-month overlapping period (in this case, October-December 1994), those results would misleadingly promote a spin-up period of only a few months. However, the divergence of the simulations during the following summer indicates that RCM applications would benefit from spin-up periods that cover a minimum of one full annual cycle so that seasonally-dependent spin-up behaviour can be excluded from the period utilized for analysis for a given application. In the present

work, spin-up criteria in Section 3d are applied over a moving 365-day window, as a shorter (e.g., monthly) window may produce a false positive result for spin-up having been achieved during the transitional and winter months. Even with this rigorous criteria, 2-m temperature and top-layer (0-10 cm) soil fields satisfy this condition within the first annual cycle for >50% of grid cells over the CONUS. On average, bottom-layer soil temperatures achieve spin-up by 11 months across the CONUS except in the Upper Midwest and Northern Rockies and Plains. Top-layer soil moisture requires a longer spin-up

time of ~21 months over the CONUS. While 4 out of the 9 regions satisfy the spin-up criteria for top-layer soil moisture within ~15 months, slower regional spin-up times of ~1.5 to 4 years are found for areas of the central CONUS. The large differences in soil conditions through the central CONUS between I94 and I74 resulted from differences in data used for initialization, which may vary in other months. However, prior spin-up studies discussed above share similar sensitivities in their methods.

Evaporation is important for spin-up, and it is the process by which I94's solution "forgets" excessive soil moisture in its initial state and converges toward the I74 simulation. The key processes by which the newly initialized model must address anomalies in the initial conditions is known to influence the timescale in which spin-up can be achieved (Cosgrove et al., 2003). The timescales for spinning up physical and hydrological processes are also considered in the modelling practice of





avoiding winter months for initialization of an RCM (as recommended by Jerez et al., 2020), as frozen soil in initial
conditions is resolved through seasonal melting. When examining WRF results over multiple domains within CORDEX,
Lavin-Gullon et al. (2023) found seasonal effects on spin-up time for soil moisture in South America, where warm season
initializations had greater uncertainty due to their coinciding with the South American rainy season, a period of increased
variability in precipitation and associated uncertainty in soil moisture. When discussing the use of splitting centennial-scale
RCM projections into shorter periods for computational efficiency, Lavin-Gullon et al. (2023) recommended increased spin-
up times and outlined an approach of utilizing three 30-year time slices with 5-year spin-up times to account for uncertainty
in the time needed for the soil fields to spin-up.

Utilizing a minimum of one full annual cycle for spin-up time excludes the re-emergence of spurious seasonally-influenced
spin-up effects that can influence key variables that are often used in RCM applications, such as seasonally and monthly
averaged 2-m temperatures. Here, regional results in the central CONUS support the use of spin-up periods of 1-4 years to
better exclude spin-up behaviour in top-layer soil moisture. Choice of the appropriate spin-up time for RCM applications
depends on several factors and often involves weighing the added computational burden against the penalty of including the
influence of spin-up within the atmospheric and land-surface fields that are most important for a given application. While
less than a year of spin-up time may appear adequate for atmospheric fields like 2-m temperature, including spurious spin-up
behaviour within other fields, such as soil moisture and evaporation, can cascade to projections of drought, heat stress, and
flooding, among others as these moisture processes affect "downstream" fields within the RCM. Here, results support a
pragmatic compromise of using at least 1 year to spin-up mid-latitude RCM simulations.

### Code Availability

The WRF model is provided by NCAR, funded by the National Science Foundation. WRF can be downloaded from
https://github.com/wrf-model/WRF. R can be downloaded from https://cran.r-project.org/.

### Data Availability

Data from the EPA Dynamically Downscaled Ensemble (EDDE) Version 1 (https://doi.org/10.23719/1530964), including
the variables discussed here, are available from Amazon Web Services Open Data Project within the repository at
https://registry.opendata.aws/epa-edde-v1/. Additional information on accessing EDDE can also be found at
https://www.epa.gov/climate-research/epa-dynamically-downscaled-ensemble-edde.

### Disclaimer

The views expressed in this article are those of the authors and do not necessarily represent the views or policies of the U.S.
EPA. Any mention of trade names, manufacturers, or products does not imply an endorsement by the United States
Government or the U.S. EPA. The U.S. EPA and its employees do not endorse any commercial products, services, or
enterprises.



**Author contribution**

MM, TS, and AJ designed the study, carried out the I74 simulation discussed here, and analysed the simulation. TS, JB and CN also contributed to developing the I94 simulation and analysis. JW provided input into the analysis. MM wrote an original draft of this publication, and all coauthors participated in review and editing.

**Competing interests**

The authors declare that they have no conflicts of interest.

**Acknowledgments**

The authors appreciate feedback provided from technical reviews of this article from Jerold Herwehe and Chunling Tang.

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
