# Peer review of "Examining Spin-Up Behaviour within WRF Dynamical Downscaling Applications"

_EGUsphere, 2025_

## Author Comment (AC1)

Response to Reviewer 1

We greatly appreciate the comments provided by the reviewer and have incorporated their feedback, as noted in each of the comments below.  The Reviewer's comments appear in black below, and our responses follow in *italicized blue text.*

Reviewer 1

Mallard et al. tackle a key practical issue in regional climate modeling: determining how long a model should be run to remove dependence on initial conditions. This is especially relevant for multi-decade simulations. The work uses a uniquely long reference run (20-year lead) to examine spin-up, extending beyond earlier studies that only looked at shorter lead times (weeks to 2 years). The results (e.g., the need for around a year or more for spin-up, particularly for soil moisture) are valuable for guiding modeling protocols and could influence future downscaling experiment design. The manuscript is well-written and logically organized. Overall, it is a very well-executed study, and I have only one minor point that I'd like addressed before publication.

*We appreciate this positive feedback on the value of the study and unique aspects of its design, as well as the comments on the organization of the manuscript.  We have incorporated the feedback the reviewer gives below to further improve the manuscript.*

That is, that the abstract and conclusion note that the results support using at least one year of spin-up. This is accurate but overemphasizes the one-year period. Since this was insufficient in some cases, I suggest clearly quantifying it in some way, in the abstract in particular - for example, acknowledging that some regions may require more.

*We highlight the need for 1 year of spin-up time as a minimum to account for seasonal effects while considering common computational limitations, but we agree that use of multi-year spin-up periods is supported by our analysis of soil conditions over regions within the central CONUS.  We appreciate the feedback that this result is underemphasized within the abstract and have revised the abstract to provide more clarity in summarizing our results.  The revised abstract ends with the following sentence:*

*Results from some regions of the CONUS indicate that spin-up durations of 1-4 years are needed to exclude spurious behavior in top-layer soil moisture, which exhibit prolonged spin-up compared with other near-surface variables examined here.*

Also, at line 34, "Georgi" should be "Giorgi", but this barely warrants comment

*This typo has been fixed.  Thank you for catching it.*

---

## Author Comment (AC2)

Response to Reviewer 1

We greatly appreciate the comments provided by the reviewer and have incorporated their feedback, as noted in each of the comments below.  The Reviewer's comments appear in black below, and our responses follow in *italicized blue text.*

Reviewer 1

Mallard et al. tackle a key practical issue in regional climate modeling: determining how long a model should be run to remove dependence on initial conditions. This is especially relevant for multi-decade simulations. The work uses a uniquely long reference run (20-year lead) to examine spin-up, extending beyond earlier studies that only looked at shorter lead times (weeks to 2 years). The results (e.g., the need for around a year or more for spin-up, particularly for soil moisture) are valuable for guiding modeling protocols and could influence future downscaling experiment design. The manuscript is well-written and logically organized. Overall, it is a very well-executed study, and I have only one minor point that I'd like addressed before publication.

*We appreciate this positive feedback on the value of the study and unique aspects of its design, as well as the comments on the organization of the manuscript.  We have incorporated the feedback the reviewer gives below to further improve the manuscript.*

That is, that the abstract and conclusion note that the results support using at least one year of spin-up. This is accurate but overemphasizes the one-year period. Since this was insufficient in some cases, I suggest clearly quantifying it in some way, in the abstract in particular - for example, acknowledging that some regions may require more.

*We highlight the need for 1 year of spin-up time as a minimum to account for seasonal effects while considering common computational limitations, but we agree that use of multi-year spin-up periods is supported by our analysis of soil conditions over regions within the central CONUS.  We appreciate the feedback that this result is underemphasized within the abstract and have revised the abstract to provide more clarity in summarizing our results.  The revised abstract ends with the following sentence:*

*Results from some regions of the CONUS indicate that spin-up durations of 1-4 years are needed to exclude spurious behavior in top-layer soil moisture, which exhibit prolonged spin-up compared with other near-surface variables examined here.*

Also, at line 34, "Georgi" should be "Giorgi", but this barely warrants comment

*This typo has been fixed.  Thank you for catching it.*

We greatly appreciate Dr. Rahimi's comments and have included additional analysis and figures into the manuscript in response, as noted in each of the comments below. The Reviewer's comments appear in black below, and our responses follow in *italicized blue text.*

Stefan Rahimi

Synopsis: The authors compare two WRF simulations, one with 20 years and another with 3 months of spin-up to assess the time scales over which two dynamically downscaled ESMs converge in terms of their soil moisture, temperature, 2-m temperature, and sensible and latent heat fluxes. They also looked at this convergence as a function of NCA region, adding another line of evidence that spin-up time is a strong function of region. While I believe that some within the regional downscaling community will not see this finding as surprising or novel, I believe that it is critical to continue revisiting questions of spin-up length, as many in the dynamical downscaling community (including me) continue to attempt to parallelize experiments across regions with smaller and smaller amounts of spin-up time to finish simulations faster. Continuous testing of spin-up is further warranted given that newer land-surface models (LSMs) continue to roll out more sophisticated and complex solvers, which may behave differently from previous generations of LSMs in terms of spin-up, potentially necessitating continuous updating of community spin-up standards. I really like the idea of giving variable-specific spin-up times in the final table of the manuscript, as it is something that folks downscaling on sub-regional scales can directly point to as a starting point in their spin-up times, and it provides a comparative benchmark to those who will contribute future studies on spin-up time length. I recommend publication of this manuscript following revisions below.

*We appreciate this feedback on the value of regionally-based results, both for those applying RCMs at regional scales and also to show how spin-up time varies across areas of the CONUS. We agree that ongoing testing of various spin-up times has the advantage of addressing changes in spin-up across updates and variations in model physics, even within the WRF community, and have included more language in the revised manuscript acknowledging that sensitivity (as detailed below). We have also expanded Table 1 in the revised manuscript to give additional information on variable-specific spin-up times, as a part of our inclusion of analysis of precipitation among the other variables already discussed in the article (as described further below).*

- Question: Why was 1994 chosen instead of 1984 or 2004? Was WY 1995 in the ESM/WRF an average precipitation year? Wet? Dry?

*1. The I94 simulations were set up with an end date of 1 Jan 2006 and an 11-year analysis period based on 1) the availability of CMIP5 global simulations for a historical period through 2005, 2) a desire to examine recent historical periods based on priorities within the Environmental Protection Agency, and 3) consideration of computational limitations. Subsequently, additional computational resources allowed us to expand to a 31-year period. The composition and expansion of the EDDE dataset, as well as the motivation behind its development, is described in detail in Spero et al. (2025), which is first referenced in the manuscript on line 77 upon introduction of the EDDE ensemble. So that this information can be more easily found, we have revised the manuscript to a more specific reference to Spero et al. (lines 101-102): "As discussed in Spero et al. (2025; their Section 2), a more prolonged historical period was later produced with a matching model configuration."*

*In order to address the reviewer's query about how to best characterize the 1995 water year, below we show a 31-year timeseries of monthly precipitation over the CONUS for I74, including the 11-year timeseries of simulated rainfall from I94 (Fig. R1). October 1994 and the proceeding water year do not appear to be anomalously wet or dry years, as compared to variability over the course of the timeseries. Although we have not examined precipitation directly from CESM for this period, both the I94 and I74 runs driven by boundary conditions and large-scale nudging from CESM simulate precipitation values for the period around initialization that appear to be well within the variability shown throughout the timeseries for the period of October 1974 through December 2005. When examining differences in precipitation over water year 1995, I94 does have ~5% more precipitation during the summer of 1995, relative to I74, when examining average seasonal precipitation for the June-July-August period. This increase in precipitation during 1995 is further discussed in the revised manuscript and in our responses below.*

[Figure]

**Fig. R1.** *Total precipitation (in mm per month) spatially averaged over land cells within the CONUS for the I74 (blue) and I94 (red) simulations shown over the period 1974-2005, with the shorter simulation only shown over the later portion of the period.*

- Line 110: What is the intensity of the spectral nudging employed here, and which relaxation and wavenumber values were used?

*2. We have included further detail in the revised manuscript on where this information can be found in a prior study.  The nudging coefficients and wavenumbers follow those used by Otte et al. (2012) within their Table 1 (excluding the analysis nudging settings, which are not employed for any runs examined in the present study).  Otte et al. is referenced in the manuscript (see line 112), and the reference has been modified to specify that these settings can be found in their Table 1.*

- 3: Maybe I missed this, but is the warmness in I94 across south-central Canada explained (Figs. 2, 3)? The following summer's relative coldness in the 2-m temperature and links to evaporative cooling are well-explained, but I couldn't seem to find this same physical argument for the preceding winter's warmness? Is this perhaps a snow difference between the two simulations?

*3. It is correct that the present study only briefly discusses the differences between I94 and I74 that fall mainly into the Canadian portion of the modeling domain (on lines 161-164).  We briefly note the warmer winter temperatures and speculate that frozen soils and snow cover may influence differences found there.  As a federal Agency, we have chosen to focus on the contiguous U.S. within this study.  However, we did examine snow cover over the entire domain within the cohort of variables we compared between I74 and I94.  When examining snow coverage over that first winter of 1994-1995, we did not find a systematic*

*decrease in snow water equivalent (SWE) over the northern areas of the domain including Canada and the CONUS (Fig. R2). Instead, a mix of positive and negative differences is present over northern areas of the domain, including throughout central Canada where the temperature differences being discussed are found (Fig. 3 within the manuscript). When averaged over all land points during the winter of 1994-1995, SWE is reduced by only 0.4 kg m$^{-2}$ (or 0.8%) in I94 relative to I74. Therefore, in our analysis focused on key differences over the CONUS, we ultimately did not include snow among the discussed variables in the manuscript.*

[Figure]

*Fig. R2*. *Monthly-averaged snow water equivalent (kg m⁻²) for I74 (left) and I94 (middle), plotted alongside the difference taken as I94 minus I74 (right) for the months of October 1994 (top), November 1994 (middle), and December 1994 (bottom).*

*The differences in soil temperature and moisture are notable over this portion of the domain falling outside the CONUS and a future study focused on processes in the cryosphere utilizing the EDDE data on Amazon Web Services Open Data Project could*

*further examine this area. However, such a study does not align with the focus on the CONUS employed within the current study due to Agency priorities.*

- Lines 249-270: It would be helpful in this table to provide each regions annual precipitation amount given the tie ins with the Cosgrove et al. (2003) finding. WRF has been updated a lot over the past several decades....do the Cosgrove et al. findings still hold up 'e.g., easier to spin up wet vs. dry soils?'

*4. The authors agree that the spin-up lengths discussed in Section 3.4 and Table 1 can be expected to be sensitive to changes in model physics, especially when considering the LSM and its effect on soil fields discussed here. We have revised the article to include a sentence further highlighting this expected sensitivity within Section 3.4 (see lines 258-260 of the revised manuscript): "Additionally, spin-up times found here can be expected to be sensitive to choice of model physics (e.g., Cosgove et al. 2003; Jankov et al., 2007; Kleczek et al., 2014; Tewari et al., 2022)."*

*While the present study does share quantitative results from Cosgrove et al. within Section 1, the conclusion from the Cosgrove study that is of most consequence to the present work is that spin-up times are influenced by the timescales of key processes needed to resolve the anomalies present in the initial fields; more specifically, it was concluded in that work that relatively wet and dry soils will be dependent on the timescales of evaporation and precipitation, respectively. The present work features generally wet soil anomalies at both the upper and mid- to bottom-layers (Figs. 3 and 4 in the manuscript). Therefore, evaporation is a key process that is heavily discussed in our results. However, we include the discussion of Cosgrove et al. in Section 3.4 to communicate to the reader our speculation that, if our analysis had found dry soil anomalies of similar magnitude in I94, spin-up times could have been even more prolonged. As we do not have an idealized simulation to test this hypothesis with, we present it as speculation and have further revised the article to properly caveat our comparison to Cosgrove et al. at lines 284-285: "However, the present work does not include an idealized simulation set up with dry soil anomalies to test this hypothesis and compare directly to results from Cosgrove et al."*

*To further address the reviewer's concerns, we have included Table R1 below with annual totals in precipitation over water year 1995. Precipitation is increased in the I94 simulation, relative to I74, by ~3% as averaged across the CONUS. Differences in the westernmost regions (Northwest, West, & Southwest) and the Northeast are less than 1%. However, regions in the central portions of the CONUS (Northern Rockies and Plains, Upper Midwest, and Ohio Valley) feature the largest differences among the simulations, with the I94 having ~4-9% more precipitation than I74 over this period.*

*Increased precipitation in this area is also consistent with Fig. 8 which was added to the revised manuscript in response to the reviewer's comment below. Generally, the contrast between I94 and I74 features generally wet soils and increased precipitation. However, an idealized simulation with similarly dry soil anomalies could be a focus of future work to better understand the role of precipitation and interactions with relatively dry soils and their effect on spin-up duration.*

| | I74 [mm] | I94 [mm] | Difference [mm] (I94-I74) | Difference (%) |
|---|---|---|---|---|
| CONUS | 864.31 | 888.32 | 24.01 | 2.78 |
| Northwest | 1351.92 | 1352.19 | 0.27 | 0.02 |
| West | 669.87 | 673.30 | 3.43 | 0.51 |
| Northern Rockies and Plains | 687.44 | 748.96 | 61.52 | 8.95 |
| Southwest | 606.86 | 603.54 | -3.32 | -0.55 |
| Upper Midwest | 849.50 | 911.36 | 61.86 | 7.28 |
| South | 660.75 | 686.94 | 26.20 | 3.96 |
| Ohio Valley | 1081.14 | 1150.71 | 69.57 | 6.44 |
| Southeast | 1224.71 | 1187.92 | -36.79 | -3.00 |
| Northeast | 1255.34 | 1266.16 | 10.83 | 0.86 |

**Table R1.** *Total precipitation [mm] over water year 1995 (1 October 1994 through 30 September 1995), spatially averaged over the CONUS and each of the 9 NCEI regions (manuscript's Fig. 1), shown alongside differences in precipitation shown in mm and as a percentage and taken as I94 minus I74.*

- 2-6: Would be nice to see the same figures but for precipitation

*5. Such a figure has been included in the revised draft as Figure 8. It, and an accompanying discussion, can be found on pages 15-16 of the revised manuscript. Generally, during the 1994 through 1995 period, precipitation is found to increase in the central and eastern portions of the CONUS for I94 relative to I74. Similar to other variables, a seasonal cycle is evident, as precipitation differences between the runs decrease in magnitude the winter months and re-emerge and increase during the following summer. As both simulations have matching boundary conditions, we assert that increased soil moisture and evaporation enhance precipitation in the central and eastern CONUS.*

*We have also revised Table 1 in the manuscript to include a quantitative examination of daily precipitation, alongside the other previously discussed variables. Our results do show that precipitation over the CONUS meets the applied spin-up criteria within the first annual window tested (days 1-364). This finding is consistent*

*with results from Jerez et al. (2020), with that study using a similar statistical test (as discussed in the first paragraph of Section 3.4) and concluding that simulated precipitation satisfied their spin-up criteria within the first week of WRF runs over a European domain. However, in the present work, it is found that some regions through the central CONUS do not meet the criteria for spin-up until 1.4-5.8 months has elapsed.  As discussed in Sections 3.3 & 3.4, as these regions also experience increased soil moisture and prolonged spin-up times for top-layer soil moisture, we hypothesize that precipitation differences are influenced by differences in the initial soil state of the runs.  Boundary conditions, and therefore long-range moisture transport, is matched between the runs and precipitation differences over the CONUS would be driven by changes in both local and upstream evaporation within the domain.*

**Examining Spin-Up Behaviour within WRF Dynamical Downscaling Applications**

[revised manuscript text omitted]

Monthly precipitation in I94 features minimal differences compared to the reference simulation over the first few months

230  following initialization, with differences in CONUS-averaged precipitation of less than 1 mm/month, followed by increased precipitation in I94 peaking at ~5 mm/month the following summer as averaged over the CONUS (Figs. 8a and 8b).  Increased precipitation is generally located though the central and eastern CONUS (Fig. 8d).  Some areas of the Southeast do show a mixed pattern of both positive and negative precipitation differences in I94, relative to I74.  However, precipitation generally increases over areas of the CONUS that experience increased latent heat release and evaporation (Fig. 6d) and increased 2-m

235  specific humidity (not shown).  As I74 and I94 feature matching boundary conditions and sea surface temperatures, enhancement of precipitation from increased moisture in the domain would be driven by differences in the initial soil state in the lower boundary condition.

[Figure]

240

**Figure 8. As in Fig. 2 but for monthly total precipitation [mm].**

**3.4 Quantitative Examination of Spin-up**

[revised manuscript text omitted]

I94 increases latent heat release and reduces sensible heat flux (Figs. 6 and 7) in response to cooler and wetter soils, following the same seasonal pattern of differences as 2-m temperature and top-layer soil temperature that are reduced in magnitude over the winter and reappeared the following summer. Meanwhile, during the initial year of the I94 simulation, deeper soil moisture values remain elevated in I94 (Fig. 5), providing additional soil moisture to drive increases the following summer when seasonally warmer temperatures support higher saturation deficits and increased evaporation. Cooler temperatures in I94 relative to I74 also re-emerge during summer 1995 (the first summer of the I94 simulation), as more near-surface evaporative cooling in the I94 run sustains cooler surface temperatures relative to I74. Precipitation increases are generally found over the central and eastern CONUS during the following summer (Fig. 8) fed by increased evaporation in I94 
[revised manuscript text omitted]